# Reliable protein–protein docking with AlphaFold, Rosetta, and replica exchange

**Ameya Harmalkar[1], Sergey Lyskov[1], Jeffrey J Gray[1,2,3]\***

[1]Department of Chemical and Biomolecular Engineering, The Johns Hopkins University, Baltimore, United States; [2]Program in Molecular Biophysics, The Johns Hopkins University, Baltimore, United States; [3]Data Science and AI Institute, Johns Hopkins University, Baltimore, United States

## eLife Assessment

The authors report how a previously published method, ReplicaDock, can be used to improve predictions from AlphaFold-multimer (AFm) for protein docking studies. The level of improvement is modest for cases where AFm is successful; for cases where AFm is not as successful, the improvement is more significant, although the accuracy of prediction is also notably lower. The evidence for the ReplicaDock approach being more predictive than AFm is particularly **convincing** for the antibody–antigen test case. Overall, the study makes a **valuable** contribution by combining data- and physics-driven approaches.

**\*For correspondence:**
jgray@jhu.edu

**Abstract** Despite the recent breakthrough of AlphaFold (AF) in the field of protein sequence-to-structure prediction, modeling protein interfaces and predicting protein complex structures remains challenging, especially when there is a significant conformational change in one or both binding partners. Prior studies have demonstrated that AF-multimer (AFm) can predict accurate protein complexes in only up to 43% of cases (Yin et al., 2022). In this work, we combine AF as a structural template generator with a physics-based replica exchange docking algorithm to better sample conformational changes. Using a curated collection of 254 available protein targets with both unbound and bound structures, we first demonstrate that AF confidence measures (pLDDT) can be repurposed for estimating protein flexibility and docking accuracy for multimers. We incorporate these metrics within our ReplicaDock 2.0 protocol to complete a robust in silico pipeline for accurate protein complex structure prediction. AlphaRED (AlphaFold-initiated Replica Exchange Docking) successfully docks failed AF predictions, including 97 failure cases in Docking Benchmark Set 5.5. AlphaRED generates CAPRI acceptable-quality or better predictions for 63% of benchmark targets. Further, on a subset of antigen-antibody targets, which is challenging for AFm (20% success rate), AlphaRED demonstrates a success rate of 43%. This new strategy demonstrates the success possible by integrating deep learning-based architectures trained on evolutionary information with physics-based enhanced sampling. The pipeline is available at https://github.com/Graylab/AlphaRED.

## Introduction

In silico protein structure prediction, that is, sequence to structure, tackles one of the core questions in structural biology. AlphaFold (AF) (*Jumper et al., 2021*) has brought a paradigm shift in the field of structural biology by intertwining deep learning (DL) tools with evolutionary data to predict single-chain structures with high accuracy. Further, AlphaFold-multimer (AFm) (*Evans et al., 2021*) and related work *Baek et al., 2021*; *Tsaban et al., 2022* have demonstrated the utility of AF to predict protein complexes. The association of proteins to form transient or stable protein complexes often involves

binding-induced conformational changes. Capturing conformational dynamics of protein–protein interactions is another grand challenge in structural biology, and many physics-based (computational) approaches have been used to tackle this challenge (*Harmalkar et al., 2022*). Computational tools have sampled the uncharted landscape of protein–protein interactions by emulating kinetic mechanisms such as conformer selection and induced-fit and identifying energetically stable binding states. However, these tools are hampered by the accuracy of the energy functions and the limitations of time and length scales for sampling. In fact, AFm predicted accurate protein complexes in only 43% of cases in one recent study (*Yin et al., 2022*). As the development of DL-based tools have unveiled ground-breaking performance in structure prediction, integration of a biophysical context has potential to strengthen prediction of protein assemblies and binding pathways.

Blind docking challenges prior to AF, particularly CASP13-CAPRI and CASP14-CAPRI experiments, reported high-quality predictions for only 8% targets (*Lensink et al., 2019*, *Lensink et al., 2021*). With the availability of AF and AFm, the CASP15-CAPRI experiment stood as its first blind assessment for the prediction of protein complexes and higher-order assemblies (*Lensink et al., 2023*). In this round, the docking community relied on AF and AFm for single-structure or complex predictions. Given that AF generates a static three-dimensional structure, it has been unclear whether conformational diversity could be captured by AF. In other terms, given a protein sequence, could AF generate ensembles of structures that include both unbound and bound conformations? Additionally, can AF reveal intrinsic conformational heterogeneity?

To diversify model complexes generated with AFm in the recent round of CASP15, predictors employed tuning parameters such as dropout (*Wallner, 2022*), higher recycles on inference (*Mirdita et al., 2022*), or modulating the MSA inputs (*Del Alamo et al., 2022*, *Wayment-Steele et al., 2022*) with the amino acid sequence. While these approaches demonstrated the ability to generate broader conformational ensembles, AFm performance still worsens with a higher degree of conformational flexibility between unbound and bound targets (*Yin et al., 2022*). Prediction accuracies especially deteriorated in bound complex regions involving loop motions, concerted motions between domains, rearrangement of secondary structures, or hinge-like domain motions, that is, large-scale conformational changes, which are also challenging for conventional docking methods (*Saldaño et al., 2022*).

Unlike state-of-the-art docking algorithms, AF's output models incorporate a residue-specific estimate of prediction accuracy. This suggests a few interesting questions: (1) Do the residue-specific estimates from AF/AFm relate to potential metrics demonstrating conformational flexibility? (2) Can AF/AFm metrics deduce information about docking accuracy? (3) Can we create a docking pipeline for in silico complex structure prediction incorporating AFm to convert sequence to structure to docked complexes?

Recent work in physics-based docking approaches tested induced-fit docking (*Harmalkar et al., 2022*), large ensembles (*Marze et al., 2018*), and fast Fourier transforms with improved energy functions (*Yan et al., 2020*) to capture conformational changes and better dock protein structures. Coupling temperature replica exchange with induced-fit docking, ReplicaDock 2.0 (*Harmalkar et al., 2022*) achieved successful local docking predictions on 80% of rigid (unbound-to-bound root mean square deviation, $RMSD_{UB}< 1.1$ Å) and 61% medium ($1.1 \leq RMSD_{UB} <2.2$Å) targets in the Docking Benchmark 5.0 set (*Vreven et al., 2015*). However, like most state-of-the-art physics-based docking methods, ReplicaDock 2.0 performance was limited for highly flexible targets: 33% success rate on targets with $RMSD_{UB} \geq 2.2$Å. Promisingly, by focusing backbone moves on known mobile residues (i.e., residues that exhibit conformational changes upon binding), ReplicaDock 2.0 sampling substantially improved the docking accuracy. But the flexible residues must first, somehow, be identified. Additionally, physics-based docking is quite slow (6–8 h on a 24-core CPU cluster) compared to recent DL-based docking tools (0.1–10 min on a single NVIDIA GPU). However, docking-specific DL tools such as EquiDock (*Ganea et al., 2021*) and dMASIF (*Sverrisson et al., 2020*) do not allow for protein flexibility, and recent tools like GeoDock (*Chu et al., 2023*) and DockGPT *McPartlon and Xu, 2023* have very limited backbone flexibility. Further, all of these DL docking tools have low success rates on unbound docking targets such as those in Docking Benchmark 5.5 (*Chu et al., 2023*).

In this work, we combine the features of a top DL approach (AFm; *Evans et al., 2021*) with physics-based docking schemes (ReplicaDock 2.0; *Harmalkar et al., 2022*) to systematically dock protein interfaces. The overarching goal is to create a one-stop, fully automated pipeline for simple, reproducible, and accurate modeling of protein complexes. We investigate the aforementioned questions and

create a protocol to resolve AFm failures and capture binding-induced conformational changes. We first assess the utility of AFm confidence metrics to detect conformational flexibility and binding site confidence. Next, we feed these metrics and the AFm-generated structural template to ReplicaDock 2.0, creating a pipeline we call AlphaRED (AlphaFold-initiated Replica Exchange Docking). We test AlphaRED's docking accuracy on a curated set of benchmark targets of bound and unbound protein structures of varying levels of binding-induced conformational change, including antibody–antigen interfaces, which additionally challenge AF2m due to the lack of evolutionary information across the interface (*Yin and Pierce, 2023*; *Ruffolo et al., 2023*). In summary, we to assess the promise of combining the best of DL and biophysical approaches for predicting challenging protein complexes.

# Results

## Dataset curation

We curated a dataset for conformational flexibility from the Docking Benchmark Set 5.5 (DB5.5) (*Vreven et al., 2015*), which comprises experimentally characterized (X-ray or cryo-EM) structures of bound protein complexes and their corresponding unbound protein subunits. Each protein target (with unbound and bound structures) is classified based on their unbound-to-bound root-mean-square-deviation ($RMSD_{UB}$) as rigid ($RMSD_{UB} \leq 1.2$ Å), medium ($1.2$ Å $< RMSD_{UB} \leq 2.2$ Å), or difficult ($RMSD_{UB} \geq 2.2$ Å). Furthermore, due to the poor performance of AF and other predictor groups in predicting antibody–antigen targets in the recent CASP15-CAPRI round *CASP15, 2022*, we identified a subset comprising only antibody–antigen complexes (including single-domain antibodies or nano-bodies) extracting all 67 antibody–antigen structures from the DB5.5 (*Guest et al., 2020*, *Vreven et al., 2015*) set. The comprehensive dataset includes 254 protein targets exhibiting binding-induced conformational changes.

For each protein target, we extracted the amino acid sequences from the bound structure and predicted a corresponding three-dimensional complex structure with the ColabFold implementation (https://github.com/YoshitakaMo/localcolabfold; *Moriwaki, 2023*) of the AlphaFold multimer v2.3.0 (released in March 2023) for the 254 benchmark targets from DB5.5. Being trained on experimentally characterized structures deposited in the PDB, AF is expected to produce models analogous to the PDB structures. Since most of the benchmark targets in DB5.5 were included in AF training, there would be training bias associated with their predictions (i.e., our measured success rates are an upper bound). However, since both unbound and bound structures exist for the benchmark targets in the PDB, we first investigated whether AFm exhibits any bias toward either unbound or bound forms for the same protein sequence. *Appendix 1—figure 1* compares the Cα-RMSD of all protein partners (calculated on a per-chain basis) of the AFm predicted complex structures from the bound (B) and unbound (U) crystal structures on a log-log scale (a few AFm predicted models were 20 Å apart from both bound and unbound structures). As evident from Supplementary Fig.S1A, the protein partners from the AFm top-ranked model deviate from both unbound and bound forms and skew more often toward the bound state. Antibody–antigen targets further demonstrate a similar trend, however with fewer targets predicted within sub-angstrom accuracy to the bound form (29.7% for Ab-Ag targets as opposed to 41% for DB5.5). We also calculated the TM-scores (*Zhang and Skolnick, 2004*) of the AFm predicted complex structures with respect to the bound and the unbound crystal structures (*Appendix 1—figure 2*). As TM-scores reflect a global comparison between structures and are less sensitive to local structural deviations, no strong conclusions could be derived. This is in agreement with our intuition that since both unbound and bound states of proteins will share a similar fold, and AF can predict structures with high TM-scores in most cases, gauging the conformational deviations with TM-scores would be inconclusive.

## AlphaFold pLDDT provides a predictive confidence measure for backbone flexibility

AF employs multiple sequence alignments with a multi-track attention-based architecture to predict three-dimensional structures of proteins and complexes. Further, for each structural prediction, it provides a residue-level confidence measure: the predicted local-distance difference test (pLDDT), estimating the agreement between predicted model to an experimental structure based on the Cα LDDT test ('Methods'). Tunyasuvunakool et al. analyzed pLDDT confidence measures for the human

proteome demonstrating the correlation between lower pLDDT scores with higher disordered regions in protein structures (*Tunyasuvunakool et al., 2021*). Building on this observation, we evaluated whether there is a correlation between AF pLDDT confidence metric and the experimental metrics of conformational change between unbound and bound structures. In this regard, we compared the computational (AF-pLDDT) and experimental (per-residue RMSD and LDDT) metrics against each other.

As a reference, we first superimposed the unbound partners over the bound structures and calculated residue-wise Cα deviations to determine the per-residue $RMSD_{BU}$ values. $LDDT_{BU}$ was measured by calculating the local distance differences in the unbound structure relative to the bound form. These metrics capture the extent of motion in the unbound–bound transitions for each of the protein targets. Next, we compared the per-residue pLDDT score from AFm predicted monomer models with the experimental metrics. *Figure 1A and B* shows the results for two representative protein targets: kinase-associated phosphatase in complex with phospho-CDK2 (1FQ1; *Song et al., 2001*) and TGF-β receptor with FKBP12 domain (1B6C; *Huse et al., 1999*). In both cases, pLDDT confidence scores correlate with the experimental measurements of binding: pLDDT decreases as $LDDT_{BU}$ decreases and $RMSD_{BU}$ increases. This is further illustrated with the AF2 predicted structures of the two targets superimposed over the bound structures (*Figure 1C*). In regions of low confidence/pLDDT (highlighted in *red*), the prediction is inaccurate, but higher confidence/pLDDT regions (highlighted in *blue*) have high accuracy of prediction with the bound form. The results for the benchmark set (*Figure 1—figure supplements 1 and 2*) show similar trends for most targets. The pLDDT, thus, can suggest protein residues that move upon binding.

## Interface-pLDDT correlates with DockQ and discriminates poorly docked structures

When the prediction accuracy is lower, it is often evident from lower confidence metrics (such as average pLDDT or PAE). However, for AFm complex predictions, the confidence metrics of the overall prediction do not correlate with the accuracy of the docked prediction, that is, even if the complex exhibits higher confidence, the docking interfaces could be incorrect. *Figure 2* shows a few examples of failed AFm predictions including rigid (2FJU [*Jezyk et al., 2006*]), medium (5VNW [*McMahon et al., 2018*]), and flexible targets (1IB1 [*Vetter et al., 1999*], 2FJG [*Fuh et al., 2006*]). In all the examples, the AFm model (highlighted in *red* to *blue* based on residue-wise pLDDT) is superimposed over an individual binding partner, and the bound structure is highlighted in *pale green*. AFm models predict the individual subunits (protein partners) accurately in almost all scenarios; however, the docking orientation is incorrect.

We investigated whether any of the AF predictive metrics could be repurposed for distinguishing native-like binding sites from non-native ones. That is, can one utilize pLDDT or PAE from AFm models to determine whether the predicted docked complex has the accurate binding orientation? Thus, we evaluated accuracy with the DockQ score, the standard metric for docking model quality (*Basu and Wallner, 2016*). DockQ $\in [0,1]$ combines interface RMSD (Irms), fraction of native-like contacts ($f_{nat}$), and ligand-RMSD (Lrms). DockQ scores above 0.23 correspond to models with a CAPRI quality of 'acceptable' or higher. As an acceptable quality target implies docked decoys are in the near-native binding region, we chose a binary classification of success with a threshold of DockQ = 0.23. We then tested how well DockQ correlated with several AFm-derived metrics: (1) interface residues: the number of interface residues (atoms of residues on one partner within 8 Å from an atom on another partner); (2) interface contacts: the number of interface contacts between the residues on the interface (Cβ atoms within 5 Å); (3) average pLDDT, determined by averaging over the per-residue LDDT score of the entire protein complex; and (4) interface-pLDDT, determined by averaging the per-residue LDDT score only over the predicted interfacial residues (as identified in case *a*).

*Figure 3A* shows the classification accuracy of each of these metrics with a receiver-operating characteristics curve. The interface-pLDDT metric stands out with a higher true positive rate (TPR) with an area under curve (AUC) of 0.86. With interface-pLDDT as a discriminating metric, we tested multiple thresholds to estimate the optimum cutoff for distinguishing near-native structures (defined as an interface-RMSD <4 Å) from the predictions. *Figure 3B* summarizes the performance with a confusion matrix for the chosen interface-pLDDT cutoff of 85. 79% of the targets are classified accurately with a precision of 75%, thereby validating the utility of interface-pLDDT as a

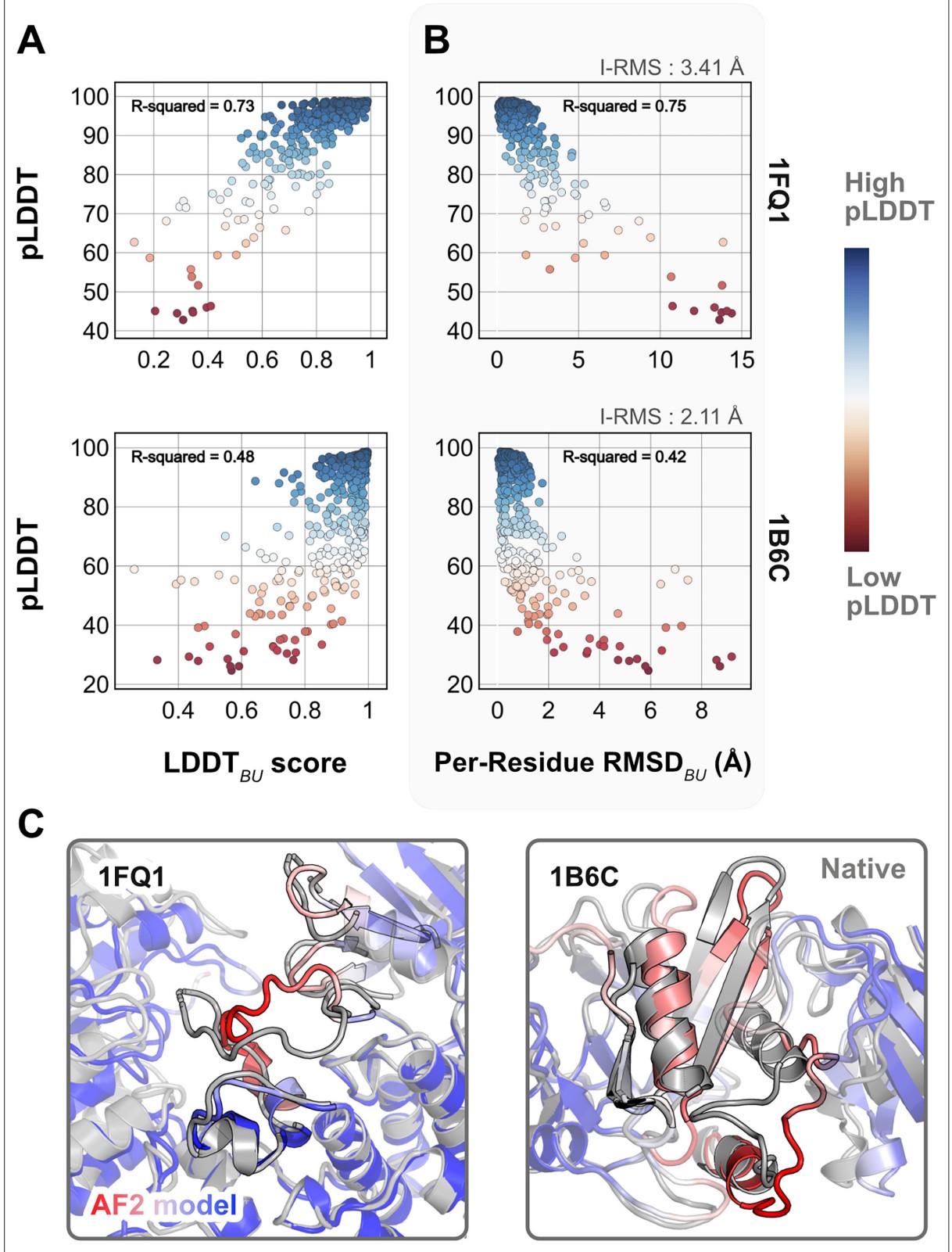

**Figure 1.** Comparison of AlphaFold-multimer (AFm) predicted local-distance difference test (pLDDT) with structural metrics. (**A**) AlphaFold pLDDT plotted against LDDT$_{BU}$. LDDT$_{BU}$ is calculated by comparing the unbound and bound environment for each residue. High scores correlate with high pLDDT (*red*). (**B**) Per-residue root-mean-square-deviation between unbound–bound structures (Per-Residue RMSD$_{BU}$) vs. AlphaFold pLDDT for two example complex structures. Higher RMSDs correlate with lower pLDDT. (**C**) Structures for two targets (PDB ID: 1B6C and 1FQ1) with the experimental

*Figure 1 continued on next page*

*Figure 1 continued*

bound form (*gray*) and the AlphaFold-multimer predicted model (*red–white–blue* in **A** and **B**). In both cases, the residues with low pLDDT scores (*red*) are the residues with incorrect conformation and more conformational change.

The online version of this article includes the following figure supplement(s) for figure 1:

**Figure supplement 1.** Comparison of AlphaFold-multimer (AFm) predicted local-distance difference test (pLDDT) with structural metrics.

**Figure supplement 2.** Comparison of AlphaFold-multimer (AFm) predicted local-distance difference test (pLDDT) with structural metrics.

discriminating metric to rank the docking quality of the AFm complex structure predictions. With newer structure prediction tools such as AlphaFold3 (*Abramson et al., 2024*) and ESM3 (*Hayes et al., 2024*) being released, investigating features that could predict flexible residues or interface site would be valuable as this information may guide local docking. This discrimination is also evident in the highlighted interface residues in *Figure 2*, where the AFm predicted models have lower confidence at predicted interfaces (*red*). Finally, we show the trend between DockQ scores and interface-pLDDT for each target in *Figure 3C*. The interface-pLDDT threshold of 85 (*dashed line*) thus can serve as the AF-derived metric to distinguish acceptable quality docked predictions from incorrect models.

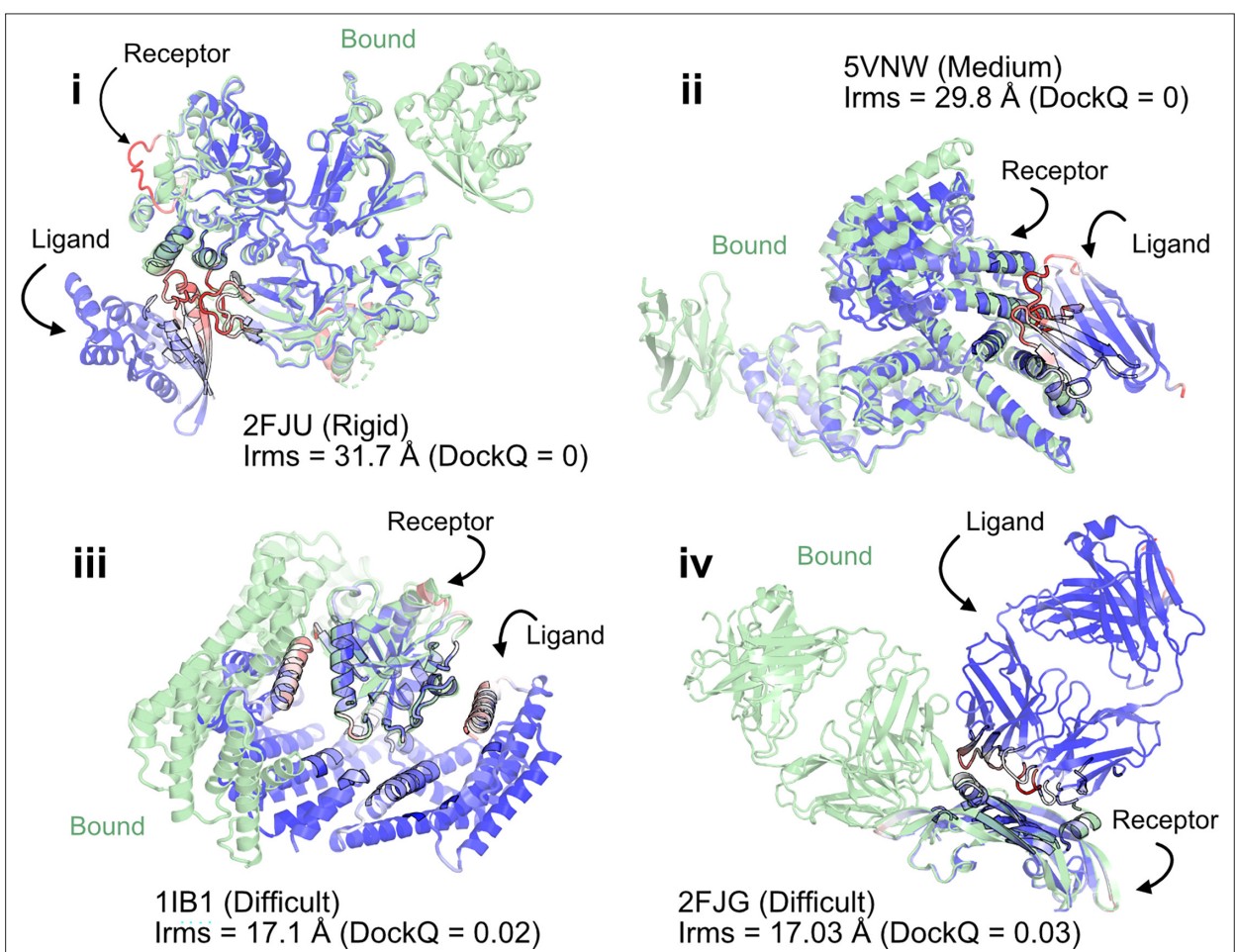

**Figure 2.** AlphaFold multimer predictions with reference to bound experimentally characterized structures. Four targets with poor DockQ scores and high interface root mean square deviations (RMSDs): (i) activated Rac1 bound to phospholipase Cβ2 (2FJU) – rigid target (RMSD$_{UB}$ = 1.04 Å); (ii) nanobody bound to serum albumin (5VNW) – medium target (RMSD$_{UB}$ = 1.49 Å); (iii) 14-3-3 zeta Isoform:serotonin N-acetyltransferase complex (1IB1) – difficult target (RMSD$_{UB}$ = 2.09 Å)l and (iv) G6 antibody in complex with the VEGF antigen – difficult target (RMSD$_{UB}$ = 2.51 Å). Bound structure in *green* and AlphaFold prediction colored by residue-wise pLDDT in *red → blue* (low confidence → high confidence).

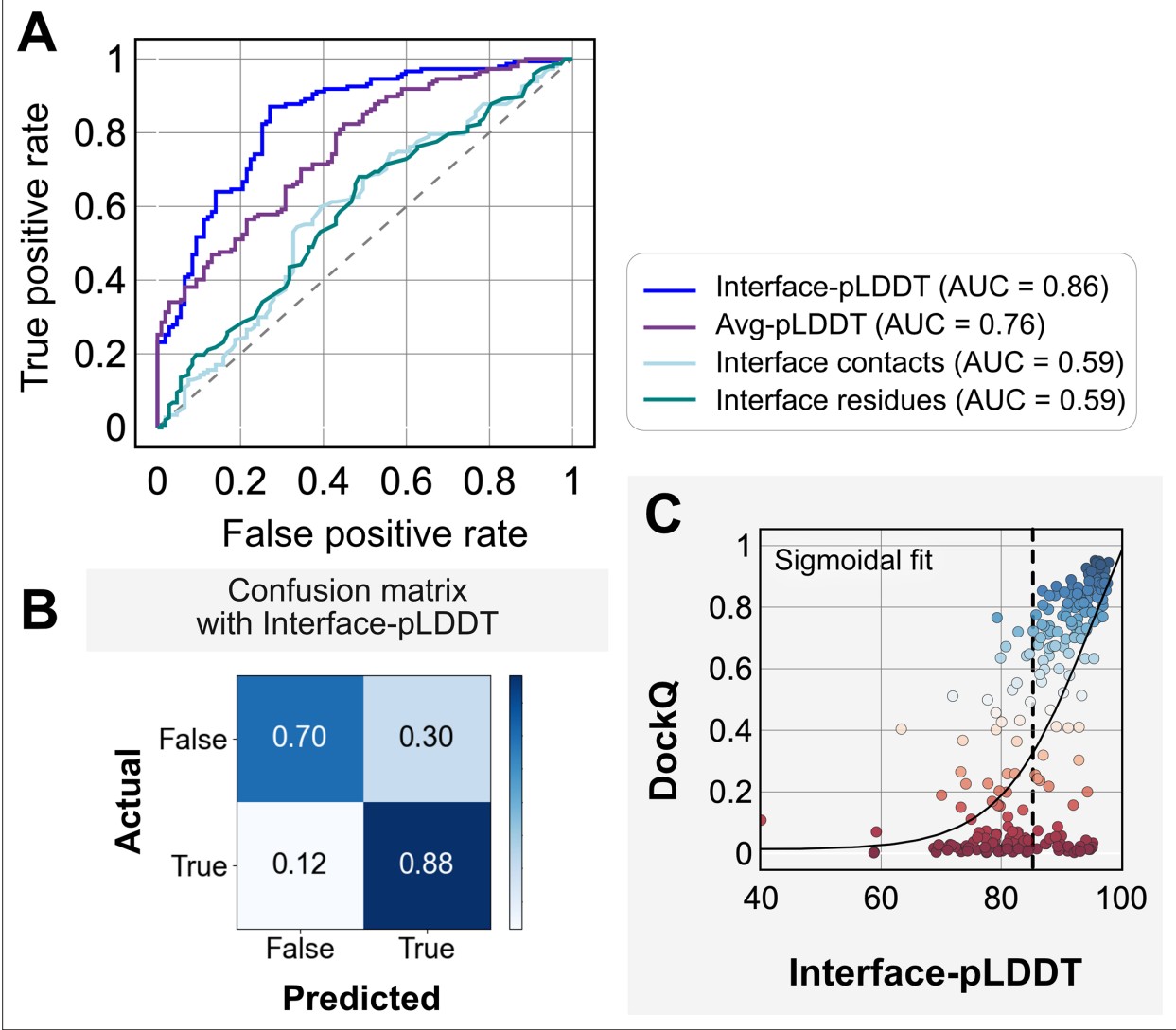

**Figure 3.** Interface predicted local-distance difference test (interface-pLDDT) is the best indicator of model docking quality. (**A**) Receiver-operator characteristics (ROC) curve as a function of different metrics for the docking dataset (n = 254). Interface residues are defined based on whether atoms of residues on one partner are within 8 Å from atom/s on another partner. Interface-pLDDT is the average pLDDT of interface residues. Avg-pLDDT corresponds to the average pLDDT across all the residues in the predicted model. Interface contacts and interface residues are the counts of the interface contacts and interface residues respectively. Interface-pLDDT has the highest area under curve (AUC) score of 0.86. (**B**) Confusion matrix with an interface-pLDDT threshold between labels predicted false (<85) and true (≥85) and an interface-RMSD threshold between labels actually true (≤4 Å) and false(>4 Å) actual labels. (**C**) Interface-pLDDT versus DockQ for all protein targets in the benchmark set. DockQ is calculated from the predicted AlphaFold structure and the experimental bound structure in the PDB. We fit a sigmoidal curve to this available data.

## Docking benchmark targets initiated from AlphaFold models improves performance

With metrics to identify the flexible regions in the protein and the docking accuracy of generated docked models, we next fused AFm with our docking protocol, ReplicaDock 2.0 (*Harmalkar et al., 2022*), to build a protocol for (1) improving on incorrect AF docking predictions and producing alternate, near-native binding models and (2) capturing backbone conformational changes with our induced-fit protocol ReplicaDock2.0 (*Harmalkar et al., 2022*). We named the protocol AlphaRED. AlphaRED uses AFm predicted structures as the primary template, estimates docking accuracy metrics, and initiates global docking or refinement protocols as required.

*Figure 4* illustrates this docking pipeline. After AFm predicts a model from the protein sequences, we calculate the interface-pLDDT to determine the docking scheme to follow. If the AFm model is

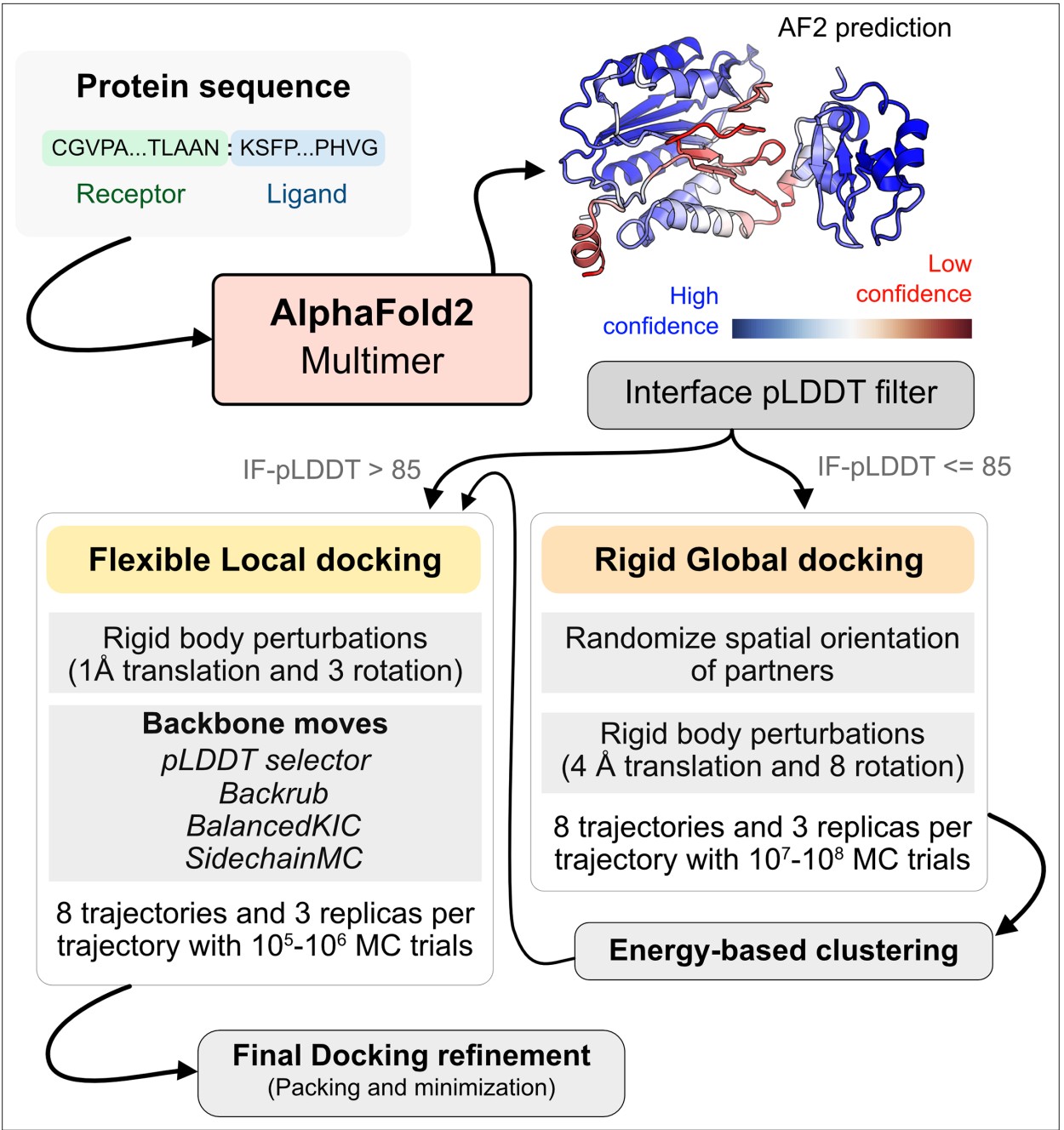

**Figure 4.** AlphaFold-initiated Replica Exchange Docking (AlphaRED) protein docking pipeline. Starting with protein sequences of putative complexes, we obtain predicted models from AlphaFold. Each model is accompanied with predicted local-distance difference test (pLDDT) scores, and based on the interface pLDDT we either initiate global rigid-body docking (interface pLDDT < 85), or flexible local docking refinement(interface pLDDT ≥ 85). For global rigid-body docking, the protein partners are first randomized in Cartesian coordinates and then docked with rigid-backbones using temperature replica exchange docking within ReplicaDock2 (**Harmalkar et al., 2022**). Decoy structures are clustered based on energy before flexible local docking refinement. In flexible local docking, we use the directed induced-fit strategy in ReplicaDock2. With mobile residues selected by the AlphaFold residue-wise pLDDT scores (threshold of 80). The protocol moves the backbones with Rosetta's Backrub or Balanced Kinematic Closure movers. Output structures are refined and top-scoring structures are selected based on interface energy.

likely to be inaccurate (interface pLDDT < 85), we initiate a global replica exchange docking simulation to explore the protein conformational landscape and identify putative binding sites. On the other hand, if the interface-pLDDT > 85 for the AFm predicted model, the docked complex is likely in the correct binding orientation. This implies the global docking stage of the protocol can be skipped

and local docking simulations can be directly initiated from the complex coordinates. Global docking follows an exhaustive, rigid-body search (no backbone moves) between the protein partners to sample putative landscapes in the energy landscape. An unbiased global docking simulation is initiated by randomizing the spatial orientation of protein partners from the input structure. The replica exchange MC routine ReplicaDock 2.0 performs rigid-body rotations (8°) and translations (4 Å). Sampled decoys are clustered from all replicas (based on energies and structural similarity) and the five top clusters are passed along for flexible local docking.

For flexible local docking, we perform aggressive backbone moves (backrub + kinematic closure, 'Methods') on candidate encounter complexes (clustered decoys), with fine rigid-body rotations and translations. To narrow conformational sampling, backbone moves are explicitly performed over residues identified as 'mobile' based on the per-residue pLDDT metric (residue pLDDT < 80). Unlike ReplicaDock 2.0 that performs induced-fit over putative interfaces, this approach targets backbone motions over these predicted mobile residues, reducing the sampling space. Local docking decoys are further refined for side-chain packing and minimization to obtain the final docked structures (details in 'Methods'). The methodological advancements and Rosetta movers in AlphaRED are further detailed in the 'Methods' section.

We investigated AlphaRED's performance on all 254 benchmark targets (*Figure 5*). Ninety-seven targets under the threshold of interface-pLDDT (≤85) were passed to the global docking branch. Targets with interface-pLDDT over 85 proceeded directly to local docking refinement. For all benchmark targets, we compared AlphaRED performance of the top-scoring decoys against initial AFm-predicted complex structures. *Figure 5A*. shows the interface-RMSD (Irms) of the AFm and AlphaRED predictions from the bound structure, respectively. The lower Irms values indicate that AlphaRED improves on existing predictions for almost all targets. For targets where AFm prediction is determined to be a failure (interface-pLDDT ≤ 85, *red*), AlphaRED demonstrates a vast improvement in Irms for 93 out of 97 targets. Additionally, for targets where AFm prediction is considered acceptable (interface-pLDDT > 85), local docking slightly improves performance. AlphaRED captures lower interface-RMSDs (under 10 Å) for targets where AFm models dock at binding sites ~40 Å away. *Figure 5B* demonstrates the improvement in recapitulating native-like contacts ($f_{nat}$) with AlphaRED.

*Figure 5C* shows the performance of the subset of antibody–antigen targets in the benchmark. Antibody targets are critical for understanding adaptive immune responses and for the design and engineering of antibody therapeutics (*Chungyoun and Gray, 2023*). However, antibodies have proven challenging for deep learning methods, especially those reliant on multiple sequence alignments, as each antibody evolves in a different organism, and their antigens evolve on a different timescale altogether. Finally, to evaluate docking success rates, we calculate DockQ for top predictions from AFm and AlphaRED, respectively (*Figure 5D*). AlphaRED demonstrates a success rate (DockQ > 0.23) of 63% for the benchmark targets. Particularly for Ab-Ag complexes, AFm predicted acceptable or better quality docked structures in only 20% of the 67 targets. In contrast, the AlphaRED pipeline succeeds in 43% of the targets, a significant improvement. Most of the improvements in the success rates are for cases where AFm predictions are worse. For targets with good AFm predictions, AlphaRED refinement results in minimal improvements in docking accuracy.

*Figure 6* highlights a global docking (a) and local docking (b) example for targets 2FJU and 5C7X, respectively. Starting from the incorrect AFm prediction (*orange*), AlphaRED samples over the conformational landscape to identify a top-scoring decoy (*blue*) with 2.6 Å Irms from the native (*gray*). *Figure 6b* shows the extent of backbone sampling with ReplicaDock 2.0 local docking. The top-scoring decoy (*blue*) samples backbone closer to the bound form, improving model quality and docking accuracy.

## Evaluation on blind CASP15 targets

All results presented thus far may be biased by the fact that these benchmark target structures were used in the AFm training. The ultimate challenge for protein structure prediction protocols is to perform successfully over blind targets such as those in CASP (Critical Assessment of protein Structure Prediction) or CAPRI (Critical Assessment of PRotein Interactions) competitions (*Kryshtafovych et al., 2021*, *Lensink et al., 2021*). CASP15 (Summer 2022) provided multiple protein docking targets (*CASP15, 2022*) that were not included in AFm training, allowing an unbiased evaluation of

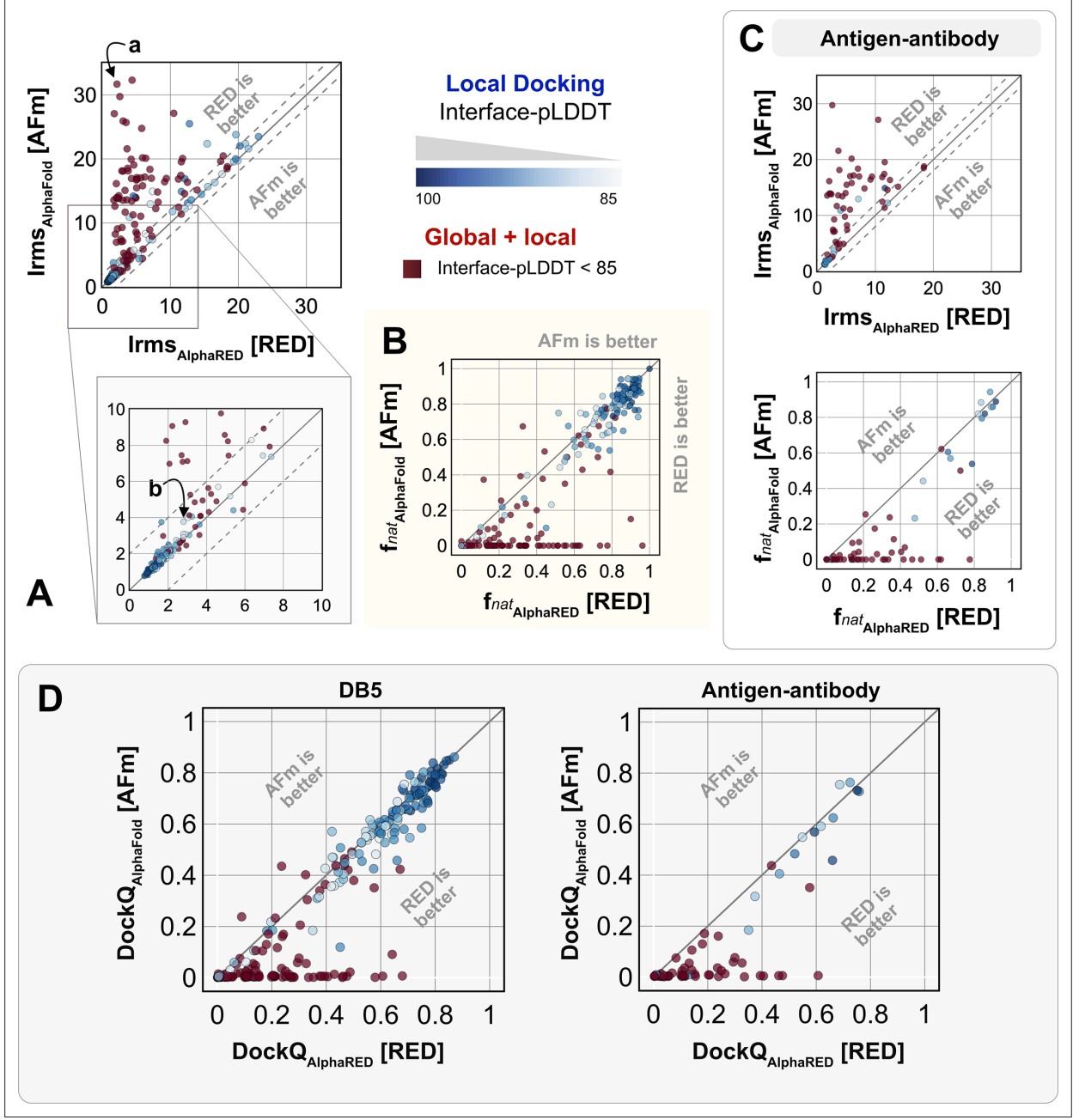

**Figure 5.** Docking performance. Targets with interface-predicted local-distance difference test (Interface-pLDDT) ≤ 85 passed first to global rigid docking (*red*) where targets with interface-pLDDT > 85 proceeded directly to local flexible backbone docking refinement (colored based on their interface-pLDDT scores; in shades of *blue*). (**A**) Interface-RMSD from AlphaFold-multimer (AFm) predicted models (y-axis) in comparison with AlphaFold-initiated Replica Exchange Docking (AlphaRED) models (x-axis). (**B**) Fraction of native-like contacts for models from AFm and AlphaRED, respectively. (**a**) and (**b**) indicate two targets (global and local docking) highlighted in *Figure 6*. (**C**) Performance on the subset of antigen–antibody targets in DB5.5. (**D**) DockQ scores for the benchmark targets (DB5) and antibody–antigen complexes.

The online version of this article includes the following source data for figure 5:

**Source data 1.** Performance of AlphaFold-initiated Replica Exchange Docking (AlphaRED) and AlphaFold-multimer (AFm) on Docking Benchmark Set 5.5.

our AlphaRED pipeline (*Lensink et al., 2023*). Thus, we tested the protocol on the five heterodimeric nanobody–antigen complexes where most of the groups performed poorly (*Figure 7*).

Since the nanobody–antigen complexes were CASP targets, we did not have unbound structures, rather only the sequences of individual chains. Therefore, for each target, we employed the AlphaRED

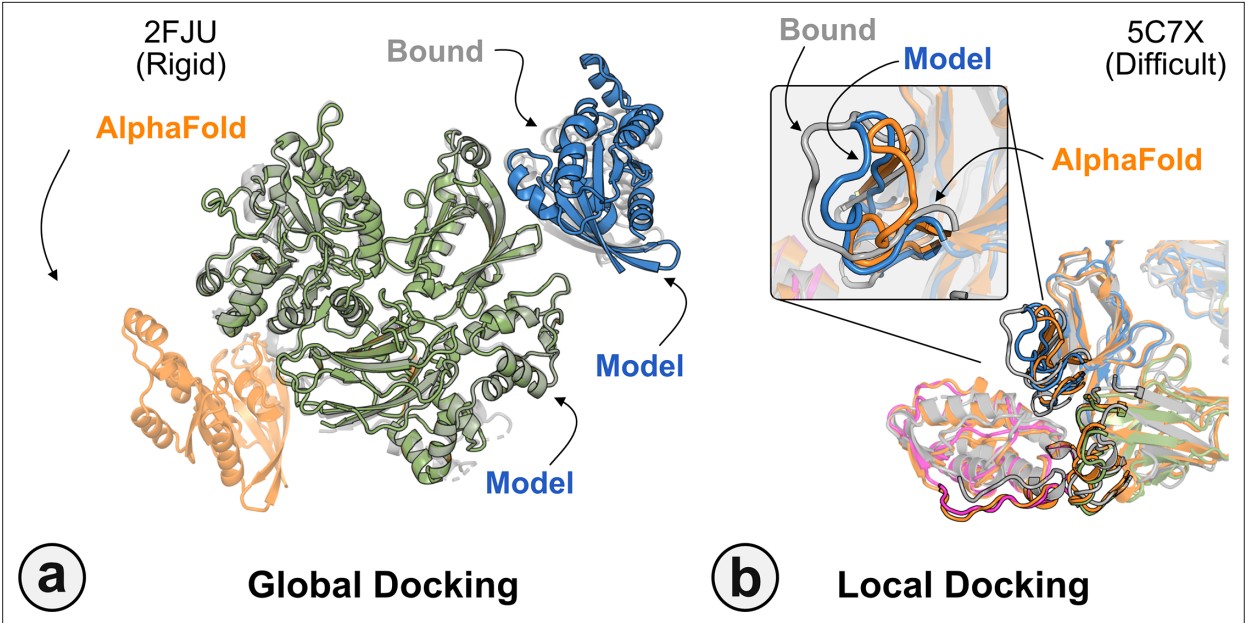

**Figure 6.** Global and local docking performance. Docking performance for targets (**a**) activated Rac1 bound to phospholipase Cβ2 (2FJU), and (**b**) neutralizing anti-human antibody Fab fragment in complex with human GM-CSF (5C7X).Starting from the AlphaFold-multimer (AFm) model (*orange*), global docking performance on 2FJU shows native-like binding site (*gray*) and sampled AlphaFold-initiated Replica Exchange Docking (AlphaRED) decoy (*blue*). For local docking, backbone sampling on mobile residues predicted by residue pLDDT (*outlined cartoon*) shows AlphaRED decoy (*blue*) moves backbone toward the bound form (*gray*).

strategy as described in *Figure 4*. All targets predicted with AFm had low interface-pLDDT, thereby demanding global docking. This is unsurprising since the targets were nanobody–antigen targets and their CDRs, particularly CDR H3, are not conserved with a scarcity of co-evolution data with the antigen (*Adolf-Bryfogle et al., 2015*). For representative target T205, our docking strategy improves the performance drastically (interface RMSD 11.4 Å for AFm model to 2.84 Å for AlphaRED) and binds in the correct site. The interface scores versus interface-RMSD plot shows a distinct funnel with low-energy medium-quality structures (*Figure 6*, top). Since the crystal structures are not yet released, the reference structure here is the top model predicted for each category in CASP15 (*Lensink et al., 2023*). For all the targets, *Figure 6*, bottom shows similar improvements for other nanobody–antigen complexes. These cases validate our strategy for blind targets and demonstrate the ability of AlphaRED to serve as a robust pipeline, integrating AF with biophysical attributes to better predict protein complex structures.

## Discussion

AF has dramatically transformed the field of structural biology and is currently the state-of-the-art method to predict protein structures from sequences, not just for monomers but also for complexes and higher assemblies (*Bryant et al., 2022*). One of the key elements of its success was the ability to mine evolutionary links between amino acids across protein families and determine structural templates. This approach dramatically improves prediction accuracy for monomers as reflected from prior CASP rounds. However, across protein interfaces, the evolutionary constraints can be weak and often skew predictions to inaccurate binding sites. Here, we demonstrated how augmenting the predictions of AF with an energy function-dependent sampling approach reveals better backbone conformational diversity and accurate prediction of protein complex structures. By utilizing the AlphaRED strategy, we show that failure cases in AFm predicted models are improved for all targets (lower Irms for 97 of 254 failed targets) with CAPRI acceptable quality or better models generated for 62% of targets overall (*Figure 8*).

First, we showed that AF confidence measures can be repurposed for estimating flexibility and docking accuracy. Interface-pLDDT, an average of the per-residue pLDDT only for the interfacial

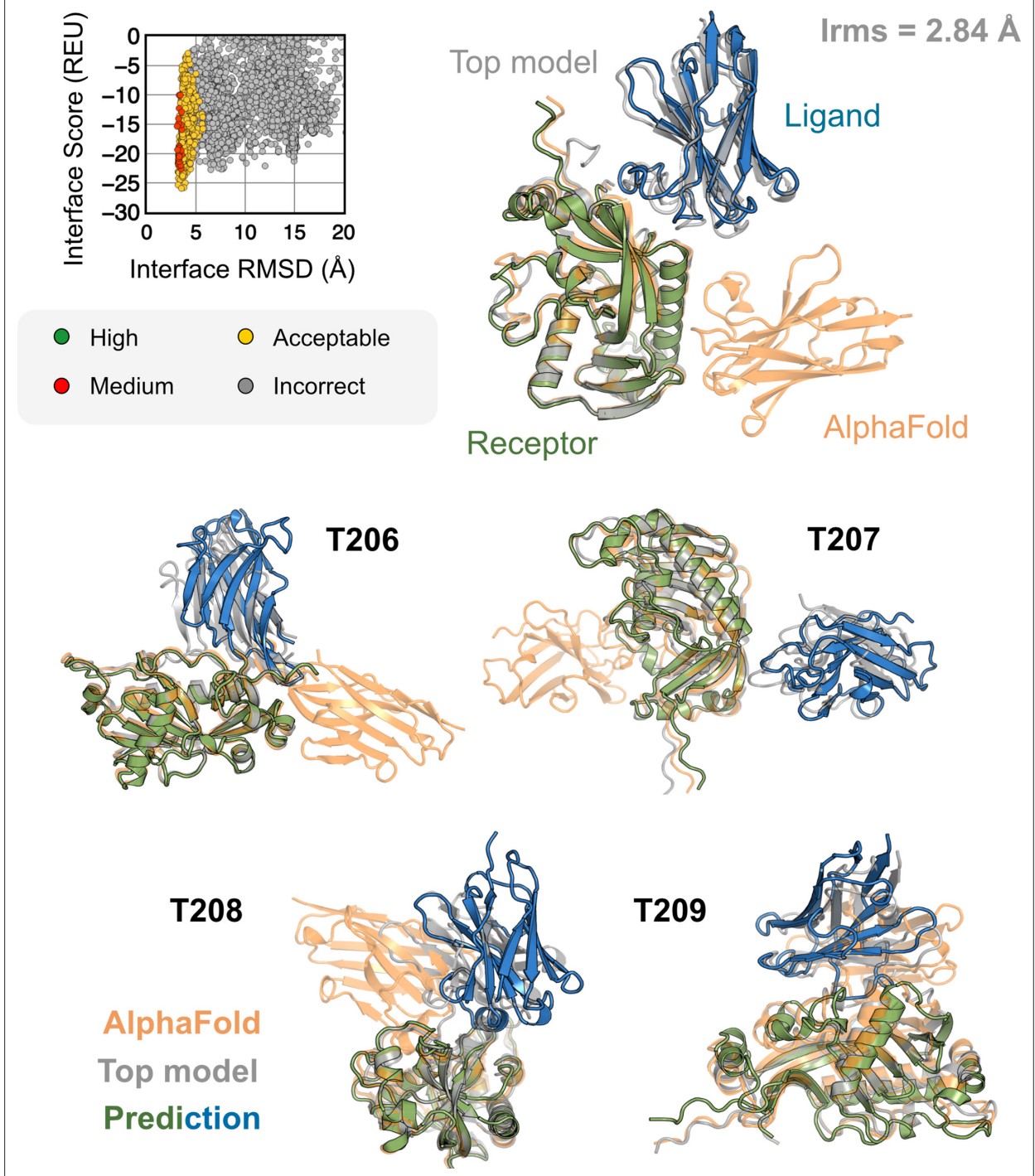

**Figure 7.** AlphaFold-multimer (AFm) and AlphaFold-initiated Replica Exchange Docking (AlphaRED) performance on CASP15 targets. Docking performance for CASP targets T205-T209. (Top) T205. Interface score (Rosetta Energy Units, REU) vs. Interface root mean square deviation (RMSD) (Å) for candidate docking structures generated by the AlphaRED docking pipeline. (Top-right) The top-scoring AlphaRED model (*green-blue*) recapitulates the native interface (*gray*) and has an interface RMSD of 2.84 Å. The distinction between the predicted model with respect to the AFm model (*orange*) is evident (bottom) Top-scoring AlphaRED predictions for targets T206, T207, T208, and T209, respectively.

residues, is a robust metric to determine whether AFm predicted binding interfaces are correct. Additionally, thresholds of per-residue pLDDT can ascertain regions of backbone flexibility upon binding. Thus, AFm predicted models can be used as input structures for ReplicaDock 2.0 guiding the choices of global or local sampling and identifying the mobile protein segments. With DL

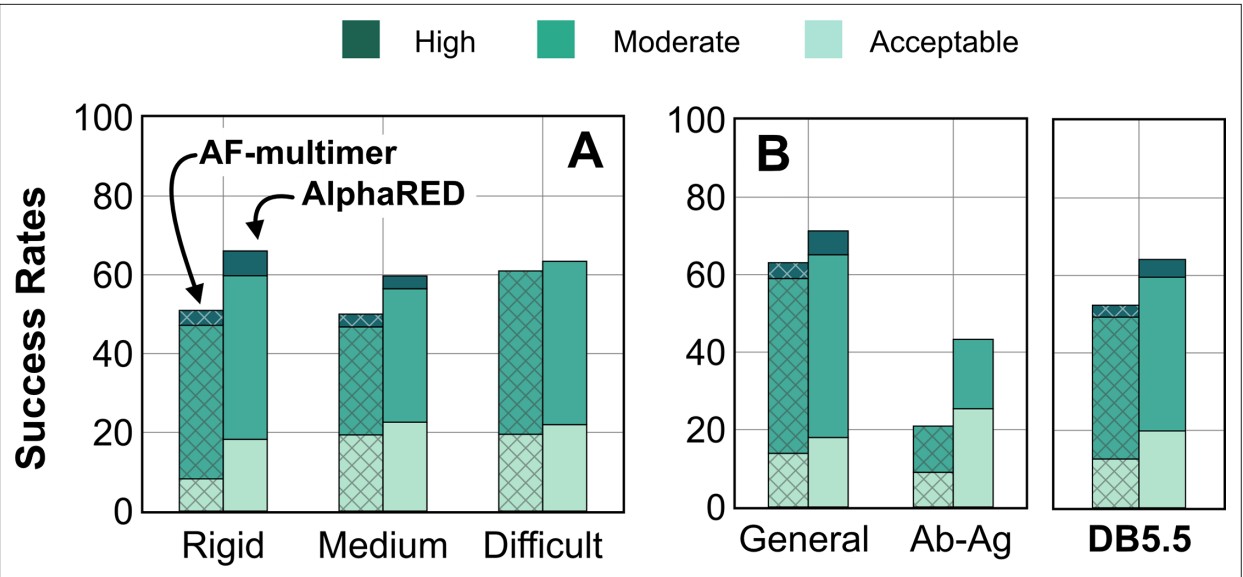

**Figure 8.** Docking prediction success with AlphaFold-multimer (AFm) and AlphaFold-initiated Replica Exchange Docking (AlphaRED). Comparison of AFm (*hashed*) and AlphaRED performance for DB5.5 benchmark set. Success rates evaluated based on DockQ criteria: *incorrect*: DockQ < 0.23; *acceptable*: DockQ ∈(0.23,0.49]; *medium*: DockQ ∈(0.49,0.8]; *high*: DockQ ≥0.8. (**A**) Classification based on the scale of flexibility: difficult (35 targets); medium (60 targets); rigid (159 targets). (**B**) Performance on the antibody–antigen complexes (67 targets) and other (non-antibody targets).

methods for structure prediction and downstream sampling with a physics-based energy function, one can efficiently explore the protein energy landscape as demonstrated with AlphaRED's performance on DB5.5. Finally, we evaluated recent CASP15 targets to investigate the extrapolation of this strategy over blind protein targets. CASP15 targets were absent from the training routine of AF and served as blind challenges to determine the efficacy of the protocol. With AlphaRED, we obtained DockQ scores over 0.23 for all five targets, with medium-quality models (DockQ > 0.49) for targets T205, T207, and T208, respectively. AFSample, a top-performing group in CASP15, employed stochastic perturbation with dropout and increased sampling to obtain medium- and high-quality models for these targets. However, AFSample requires GPU simulations to produce ~240× models with compute time ~1000× more than the baseline AFm (***Wallner, 2022***). On other hand, we utilized ColabFold (***Mirdita et al., 2022***) to generate 1–5 structures for our docking routine with the baseline version. As opposed to a couple of days on GPU (each GPU node contains up to 48 cores) utilized by AFSample, our docking routine fused with ColabFold uses 5–7 h on our CPU cluster (runs on 1 node, with 24 cores, approximating to ~100 h of CPU-hours per target). The AlphaRED docking strategy demonstrates a new and better way to predict protein complex structures within feasible compute times.

This work is particularly impactful for its success rate on antibody–antigen targets. DL promises accurate design and optimization of antibody therapeutics (***Chungyoun and Gray, 2023***), but a lack of fast and accurate docking methods for antibodies prevent high-throughput computational screening. Additionally, this work is impactful because by integrating a physics-based method for refinement, the pipeline can potentially handle post-translationally modified proteins or non-canonical residue types that are not defined in ML approaches like AF.

With this work, we have built upon the recent advances in structural biology to develop a robust tool for protein docking. We fused DL tools with conventional physics-based sampling tools to develop a pipeline that extracts the best outcomes of each methodology, where DL methods generate accurate, static structures, and physics-based sampling provides diversity and better discrimination. The protein conformational landscape is vast and DL tools such as AF provide a snapshot of relevant local minima that can aid in narrowing down the degrees of freedom in sampling (***Roney and Ovchinnikov, 2022***). With the paradigm shift in computational structural biology toward DL approaches, integrating physics within these models has tremendous potential toward understanding protein dynamics, modulating protein–protein interactions, and downstream applications to protein design.

## Methods

### Prediction of structures

For each target in the DB5.5 dataset, we first extracted the corresponding FASTA sequence for the bound complex and then obtained AF predicted models with the ColabFold v1.5.2 (*Ovchinnikov, 2021*) implementation of AF (*Jumper et al., 2021*) and AlphaFold-multimer (v.2.3.0) (*Evans et al., 2021*). Each prediction run was performed without templates, with automatic alignments and the default number of recycles to generate five relaxed predictions. Each AF prediction includes a per-residue pLDDT (predicted LDDT) measurement (*Mariani et al., 2013*), a confidence measure in prediction accuracy, and predicted template alignment (pTM) score (*Zhang and Skolnick, 2004*). The models were structurally compared with the unbound and bound structures (deposited in the PDB) for measuring flexibility, similarity, and accuracy of docking prediction.

### Metrics for backbone flexibility: RMSD and LDDT

Structures of proteins deposited in the PDB (*Berman et al., 2000*) provide a static representation of the native state of the protein. However, structural diversity has been captured by experimental techniques to identify different states of a protein in diverse physiological or chemical states, for example, catalysis (*Kingsley and Lill, 2015*), transport (*Gora et al., 2013*), and ligand binding (*Gunasekaran and Nussinov, 2007*). For protein docking challenges in particular, conformational changes are binding-induced, leading to structural differences between unbound and bound structures of protein targets.

To measure the conformational change in protein structures, we calculated two metrics: Cα RMSD and LDDT (*Mariani et al., 2013*). To get a detailed representation of the intrinsic motion of a protein, we calculated RMSDs at a residue level, that is, per-residue Cα RMSD for each residue of a protein target. The sequences + structures of unbound and bound proteins were aligned and the RMSDs were calculated for the aligned residues. The total sequence lengths were also matched and lingering end-termini residues were trimmed to ensure structural and sequential similarity.

LDDT is a superimposition-free score that estimates local distance differences in a model relative to a reference structure (*Mariani et al., 2013*). Unlike the global distance test (*Zemla, 2003*) score based on rigid-body superimposition, the LDDT score measures the conserved local interactions in the protein model to the reference. For every residue, it computes the distance between all pair of atoms $D(i,j)$ in both the model and the reference structure (bound) within a threshold (defined as the inclusion radius, generally set to 10 Å). For each pairwise distance in both distance vectors, if the distance is within the threshold, the distance is considered conserved and the fraction of conserved distances is calculated. The final LDDT score is the average of this fraction for the tolerances of 0.5, 1, 2, and 4 Å.

For a protein structure with $N$ number of residues, the overall LDDT score can be given as follows:

$$\text{dists\_to\_score}(i,j) \tag{1}$$

where norm is the normalization factor

$$\text{norm} = \frac{1}{\sum_{i,j} \text{dists\_to\_score}(i,j)} \tag{2}$$

and $\text{score}(i,j)$ is the LDDT score for the residue $i$ with respect to every other residue $j$

$$\text{score}(i,j) = 0.25 \cdot \Big\{ \, bool[\Delta D(i,j) < 0.5] +$$
$$bool[\Delta D(i,j) < 1.0] +$$
$$bool[\Delta D(i,j) < 2.0] +$$
$$bool[\Delta D(i,j) < 4.0] \, \Big\}$$

$\Delta D(i,j)$ denotes the absolute difference between $D_{\text{true}}(i,j)$ and $D_{\text{predicted}}(i,j)$ calculated as follows:

$$\Delta D(i,j) = |D_{\text{true}}(i,j) - D_{\text{predicted}}(i,j)| \tag{3}$$

where $D_{true}(i,j)$ and $D_{predicted}(i,j)$ denote the distances between the Cα coordinates of the $i$th residue and the $j$th residue for the true (reference) and predicted (model) structures, respectively. Let $x_i^k$ and $y_i^k$ represent the $k$th coordinate of the Cα atom in the $i$th residue in the reference (true) structure and predicted structure respectively, such that

$$D_{true}(i,j) = \sqrt{\sum_{k=1}^{3} \left(x_i^k - x_j^k\right)^2} \text{ and } D_{predicted}(i,j) = \sqrt{\sum_{k=1}^{3} \left(y_i^k - y_j^k\right)^2} \tag{4}$$

Finally, the distances to score (dists_to_score$(i,j)$) are computed as those pairwise distances within an inclusion radius (cutoff = 10 Å). $m_i^j$ is the mask value (1 or 0) indicating if the $j$th coordinate of the Cα atom in the $i$th residue exists in the true structure.

$$dist\_to\_score(i,j) = \begin{cases} 1 \text{ if } D_{true}(i,j) < \text{cutoff} \cdot m_i^j \cdot m_j^i \cdot (1 - \delta_j N) \\ 0 \text{ otherwise} \end{cases} \tag{5}$$

where $\delta$ = Kronecker Delta.

The advantage of the LDDT measurement lies in the estimation of relative domain orientations in multi-domain proteins or concerted motions (e.g., hinge-like moves in closed and apo proteins). In these cases, the RMSDs would be relatively high for all residues in the mobile domain; however, since the inter-residue distances within the domains are conserved, they would provide an inaccurate depiction of flexibility for the protein. Estimating both RMSDs and LDDT scores allows us to obtain a nuanced perspective of flexibility during protein association based on experimental structures.

## Developing a pipeline for protein docking

Using AlphaFold2 as a structural module, we built a pipeline for protein–protein docking to better predict protein complex structures with relatively higher accuracy. As illustrated in *Figure 4*, given a sequence of a protein complex, we use the ColabFold implementation of AF2-multimer to obtain a predictive template. An interface-pLDDT filter determines the accuracy of the docking prediction of the top-ranked model from AFm. If the interface-pLDDT ≤ 85, the prediction has lower confidence in the docking orientation, and the protocol initiates a rigid, global docking search with ReplicaDock 2.0. Implementation of ReplicaDock 2.0 (global docking) is similar to the version reported in prior work (*Harmalkar et al., 2022*). Each simulation initiates eight trajectories across three temperature replicas with inverse temperatures set to $1.5^{-1}$ kcal$^{-1}$.mol, $3^{-1}$ kcal$^{-1}$.mol, and $5^{-1}$ kcal$^{-1}$.mol, respectively. Across each replica within each trajectory, rigid-body perturbations (4 Å translations and 8° rotations) are performed for an exhaustive global search. Next, we perform an energy-based clustering of the models to obtain diverse and energetically favorable clusters. Five cluster centers (decoys) are selected and passed to the flexible local docking stage to sample conformational changes.

On the other hand, if the interface-pLDDT > 85, the binding orientation has higher confidence and the protocol directly performs a flexible local docking simulation skipping the rigid, global docking. In this stage, we perform smaller rigid-body perturbations (1 Å translations and 3° rotations) and aggressive backbone moves using a set of backbone and side-chain movers: Rosetta Backrub (*Smith and Kortemme, 2008*), Balanced Kinematic Closure (BalancedKIC), and Sidechain. The sampling weights are biased such that backbone and side-chain movers are weighted higher than rigid-body moves (3:1 weightage for backbone:rigid-body moves). We perform directed backbone sampling by focusing on predicted mobile residues (per-residue pLDDT < 80). This is automated with the BFactorResidueSelector that selects contiguous sets of residues below the specified pLDDT threshold.

However, unlike the induced-fit strategy in ReplicaDock (*Harmalkar et al., 2022*), we perform backbone sampling directed only on the mobile residues (with per-residue pLDDT < 80) identified from the AF model. We automate it using the BFactorResidueSelector to select contiguous sets of residues below the specified pLDDT threshold in the prior section. This residue subset is passed along to the backbone movers to sample backbone moves along with small rigid-body moves. Sampled decoyed are then refined, that is, undergo side-chain packing and minimization, to output docked decoys. The best ranked decoys based on interface scores are then identified as the top-scoring structures.

## Acknowledgements

The authors thank Sergey Ovchinnikov and Yoshitaka Moriwaki for ColabFold implementation of AlphaFold.This work was supported by the National Institute of Health through grant R35-GM141881 (all authors).

## Additional information

### Competing interests

Jeffrey J Gray: JJG is an unpaid board member (co-director) of the Rosetta Commons. Under institutional participation agreements between the University of Washington, acting on behalf of the Rosetta Commons, Johns Hopkins University may be entitled to a portion of revenue received on licensing Rosetta software including some methods described in this paper. JJG has a financial interest in Cyrus Biotechnology. Cyrus Biotechnology distributes the Rosetta software, which may include methods described in this paper. These arrangements have been reviewed and approved by the Johns Hopkins University in accordance with its conflict-of-interest policies. The other authors declare that no competing interests exist.

### Funding

| Funder | Grant reference number | Author |
| --- | --- | --- |
| National Institutes of Health | R35-GM141881 | Ameya Harmalkar<br>Sergey Lyskov<br>Jeffrey J Gray |

The funders had no role in study design, data collection and interpretation, or the decision to submit the work for publication.

### Author contributions

Ameya Harmalkar, Conceptualization, Data curation, Formal analysis, Investigation, Methodology, Writing – original draft, Writing – review and editing; Sergey Lyskov, Software, Methodology, Writing – review and editing; Jeffrey J Gray, Conceptualization, Resources, Supervision, Project administration, Writing – review and editing

### Author ORCIDs

Ameya Harmalkar ⬤ https://orcid.org/0000-0001-6863-9634
Sergey Lyskov ⬤ https://orcid.org/0000-0001-6380-6712
Jeffrey J Gray ⬤ https://orcid.org/0000-0001-6380-2324

Reviewer #1 (Public review): https://doi.org/10.7554/eLife.94029.3.sa1
Reviewer #2 (Public review): https://doi.org/10.7554/eLife.94029.3.sa2
Author response https://doi.org/10.7554/eLife.94029.3.sa3

## Additional files

### Supplementary files
MDAR checklist

### Data availability

AlphaRED utilizes ColabFold for structure prediction with Rosetta-based docking. The source code for AlphaRED is available on GitHub (https://github.com/Graylab/AlphaRED; copy archived at *Lyskov and Harmalkar, 2025*). To ensure ease of availabilty for researchers, we have implemented an online server on the Gray Lab ROSIE server (rosie.graylab.jhu.edu). Users can submit their prediction and docking jobs on https://r2.graylab.jhu.edu/apps/submit/alpha-red. The server would implement the AlphaRED pipeline and for input sequences would provide docked models with Rosetta energies for

further analysis. We expect this implementation to be a great resource for modeling and better understanding protein-protein interactions.

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

## Appendix 1

### Supplementary methods

#### Benchmark sets

The benchmark set was curated from Docking Benchmark 5.5 (*Vreven et al., 2015*), with targets classified based on their extent of flexibility, that is, rigid, medium, and difficult. We also curated a subset of only antigen–antibody/nanobody targets from the overall set. For each target in the benchmark set, a FASTA sequence was obtained with individual chains separated by a colon (:) indicating chain break. This sequence was used for AFm structure prediction. For target 1N2C, AFm could not generate a structural prediction owing to the longer sequence length and is excluded from the benchmark. For each structural prediction that was generated, comparisons were made to its corresponding bound and unbound forms as obtained from the Docking Benchmark 5.5.

#### Metrics and evaluation

The docking performance was evaluated based on interface RMSD (Irms), fraction of native-like contacts ($f_{nat}$), CAPRI quality, and DockQ scores. These metrics are defined as follows:

> Interface RMSD (Irms): The RMSD of all atoms on the interface in a docked protein structure relative to a reference structure (native). Interface residues are defined as all 24 amino acid residues within 10 Å of any residue on the binding partner.
> Fraction of native-like contacts ($f_{nat}$): The fraction of native-like contacts recovered in the docked structure relative to the reference structure (native).
> CAPRI quality: A CAPRI-based rank calculated on the basis of I-rms, $f_{nat}$, and ligand-RMSD to classify a docked model as incorrect, acceptable, medium, or high-quality prediction.
> DockQ Score: Similar to CAPRI quality, the DockQ scores estimates a score ($\in[0,1]$) estimating the accuracy of the docked complexes. We calculated this score based on the methodology described in *Basu and Wallner, 2016*.
> Interface score (Isc): The interface score is analogous to thermodynamic binding energy of protein association. This score is estimated by calculating the total score (Gibbs free energy) of protein complex and then by subtracting individual (monomeric) scores of protein partners in absence of its partner. Mathematically, for proteins A and B forming a complex AB, it can be defined as follows:

$$\Delta\Delta G_{\text{interface}} = \Delta G_{\text{AB}} - \Delta G_{\text{A}} - \Delta G_{\text{B}}$$

## Supplementary results

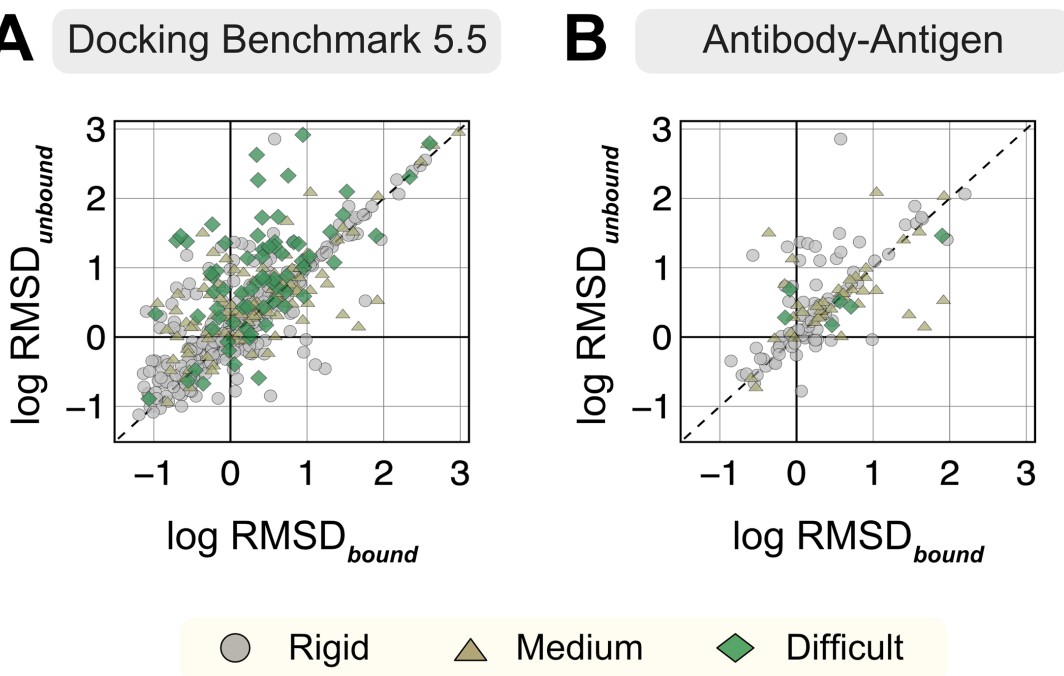

**Appendix 1—figure 1.** Root mean square deviations (RMSDs) of AlphaFold-multimer structures from experimental unbound and bound structures. Distribution of the RMSD between the AlphaFold-multimer prediction top-ranked model and the experimental unbound and bound structures. For each target, the protein partners are split into receptor and ligand respectively for comparison. Each symbol represents a category of flexibility (rigid, medium, and flexible). (**A**) Dockground Benchmark set 5.5; (**B**) antibody/nanobody–antigen targets from the benchmark.

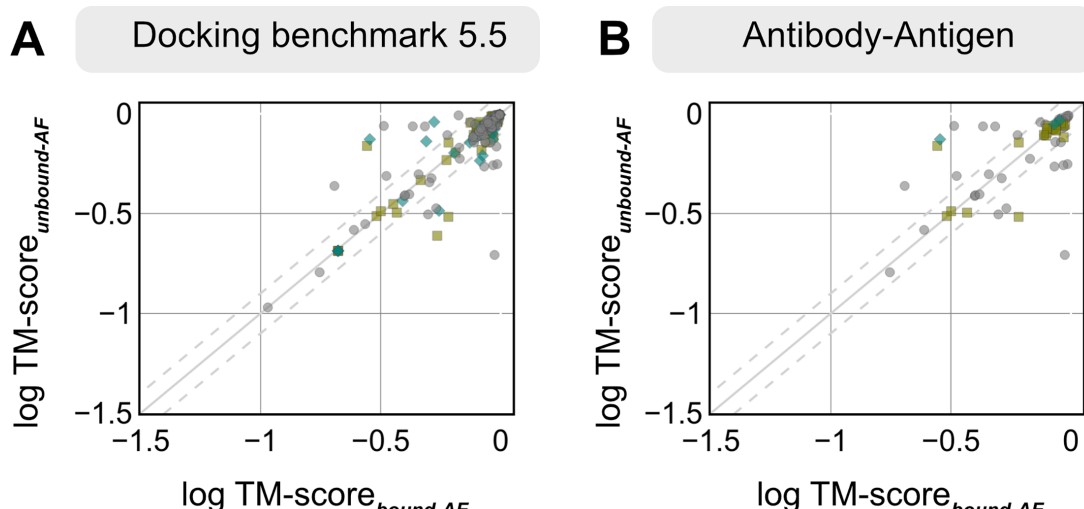

**Appendix 1—figure 2.** TM-scores of AlphaFold-multimer structures from experimental unbound and bound structures. Distribution of the TM-score between the AlphaFold-multimer prediction top-ranked model and the experimental unbound and bound structures. For each target, the protein partners are split into receptor and ligand respectively for comparison. Each symbol represents a category of flexibility (rigid, medium, and flexible). (**A**) Dockground Benchmark set 5.5; (**B**) antibody/nanobody–antigen targets from the benchmark.

