## [Editor Report · eLife Assessment]

The authors report how a previously published method, ReplicaDock, can be used to improve predictions from AlphaFold-multimer (AFm) for protein docking studies. The level of improvement is modest for cases where AFm is successful; for cases where AFm is not as successful, the improvement is more significant, although the accuracy of prediction is also notably lower. The evidence for the ReplicaDock approach being more predictive than AFm is particularly **convincing** for the antibody–antigen test case. Overall, the study makes a **valuable** contribution by combining data- and physics-driven approaches.

---

## [Referee Report · Reviewer #1 (Public review)]

Summary:

The authors wanted to use AlphaFold-multimer (AFm) predictions to reduce the challenge of physics-based protein-protein docking.

Strengths:

They found two features of AFm predictions that are very useful. (1) pLLDT is predictive of flexible residues, which they could target for conformational sampling during docking; (2) the interface-pLLDT score is predictive of the quality of AFm predictions, which allows the authors to decide whether to do local or global docking.

Weaknesses:

(1) As admitted by the authors, the AFm predictions for the main dataset are undoubtedly biased because these structures were used for AFm training. Could the authors find a way to assess the extent of this bias?

(2) For the CASP15 targets where this bias is absent, the presentation was very brief. In particular, I'm interested in seeing how AFm helped with the docking? They may even want to do a direct comparison with docking results w/o the help of AFm.

Comments on revisions:

This revision has addressed my previous comments.

---

## [Referee Report · Reviewer #2 (Public review)]

Summary:

In short, this paper uses a previously published method, ReplicaDock to improve predictions from AlphaFold-multimer. The method generated about 25% more acceptable predictions than AFm, but more important is improving an Antibody-antigen set, where more than 50% of the models become improved.

When looking at the results in more detail, it is clear that for the models where the AFm models are good, the improvement is modest (or not at all). See, for instance, the blue dots in Fig 6. However, in the cases where AFm fails, the improvement is substantial (red dots in Fig 6), but no models reach a very high accuracy (Fnat ~0.5 compared to 0.8 for the good AFm models). So the paper could be summarized by claiming, "We apply ReplicaDock when AFm fails", instead of trying to sell the paper as an utterly novel pipeline. I must also say that I am surprised by the excellent performance of ReplicaDock - it seems to be a significant step ahead of other (not AlphaFold) docking methods, and from reading the original paper, that was unclear. Having a better benchmark of it alone (without AFm) would be very interesting.

These results also highlight several questions I try to describe in the weakness section below. In short, they boil down to the fact that the authors must show how good/bad ReplicaDock is at all targets not only the ones where AFm fails. In addition, I have several more technical comments.

Strengths:

Impressive increase in performance on AB-AG set (although a small set and no proteins).

Weaknesses:

The presentation is a bit hard to follow. The authors mix several measures (Fnat, iRMS, RMSDbound, etc). In addition, it is not always clear what is shown. For instance, in Fig 1, is the RMSD calculated for a single chain or the entire protein? I would suggest that the author replace all these measures with two: TM-score when evaluating the quality of a single chain and DockQ when evaluating the results for docking. This would provide a clearer picture of the performance. This applies to most figures and tables. For instance, Fig 9 could be shown as a distribution of DockQ scores.

The improvements on the models where AFm is good are minimal (if at all), and it is unclear how global docking would perform on these targets, nor exactly why the plDDT<0.85 cutoff was chosen. To better understand the performance of ReplicaDock, the authors should therefore (i) run global and local docking on all targets and report the results, (ii) report the results if AlphaFold (not multimer) models of the chains were used as input to ReplicaDock (I would assume it is similar). These models can be downloaded from AlphaFoldDB.

Further, it would be interesting to see if ReplicaDock could be combined with AFsample (or any other model to generate structural diversity) to improve performance further.

The estimates of computing costs for the AFsample are incorrect (check what is presented in their paper). What are the computational costs for RepliaDock global docking?

It is unclear strictly what sequences were used as input to the modelling. The authors should use full-length UniProt sequences if that were not done.

The antibody-antigen dataset is small. It could easily be expanded to thousands of proteins. It would be interesting to know the performance of ReplicaDock on a more extensive set of Antibodies and nanobodies.

Using pLDDT on the interface region to identify good/bas models is likely suboptimal. It was acceptable (as a part of the score) for AlphaFold-2.0 (monomer), but AFm behaves differently. Here, AFm provides a direct score to evaluate the quality of the interaction (ipTM or Ranking Confidence). The authors should use these to separate good/bad models (for global/local docking), or at least show that these scores are less good than the one they used.

Comments on revisions:

The inclusion of the DockQ improved the paper. No further comments.

---

## [Author Response]

The following is the authors’ response to the original reviews.

**Reviewer #1 (Public Review)**
Summary:The authors wanted to use AlphaFold-multimer (AFm) predictions to reduce the challenge of physics-based protein-protein docking.Strengths:They found that two features of AFm predictions are very useful. (1) pLLDT is predictive of flexible residues, which they could target for conformational sampling during docking; (2) the interface-pLLDT score is predictive of the quality of AFm predictions, which allows the authors to decide whether to do local or global docking.Weaknesses:(1) As admitted by the authors, the AFm predictions for the main dataset are undoubtedly biased because these structures were used for AFm training. Could the authors find a way to assess the extent of this bias?

Indeed, the AFm training included most of the structures in the DB5 benchmark for its training as many structures (either unbound or bound) were deposited before the training cut-off period. One of the challenges of estimating this bias is the availability of new structures - both bound and unbound deposited after the training cut-off. Estimating the extent of training bias is therefore conditional on these factors and difficult. A few studies have attempted to address this bias (Yin et al, 2022, https://doi.org/10.1002/pro.4379).

In our study, we assess this bias by comparing the AFm structures to the bound and unbound forms and calculating their Ca RMSDs and TM-scores (new addition). We now elaborate in the Results:Dataset curation section and we have added a figure comparing the TM-scores in the supplement.

We added a clarifying text and a note about the TM-score calculation in the manuscript as follows:

“Since most of the benchmark targets in DB5.5 were included in AlphaFold training, there would be training bias associated with their predictions (i.e. our measured success rates are an upper bound).”

“We also calculated the TM-scores of the AFm predicted complex structures with respect to the bound and the unbound crystal structures (Supplementary Figure S2). As TM-scores reflect a global comparison between structures and are less sensitive to local structural deviations, no strong conclusions could be derived. This is in agreement with our intuition that since both unbound and bound states of proteins will share a similar fold, and AlphaFold can predict structures with high TM-scores in most cases, gauging the conformational deviations with TM-scores would be inconclusive.”

(2) For the CASP15 targets where this bias is absent, the presentation was very brief. In particular, it would be interesting to see how AFm helped with the docking. The authors may even want to do a direct comparison with docking results without the help of AFm.

Unfortunately since this was a CASP-CAPRI round, the structure of the unbound Antigen or the nanobodies was unavailable. Thus we cannot perform a comparison without using AF2 at all since we need a structure prediction tool to produce the unbound nanobody and the nanobody-antigen complex template structure to dock. This has been clarified in the main text for better understanding for the readers.

“Since the nanobody-antigen complexes were CASP targets, we did not have unbound structures, rather only the sequences of individual chains. Therefore, for each target, we employed the AlphaRED strategy as described in Fig 7.”

**Reviewer #1 (Recommendations For The Authors):**
For suggestions for major improvements, see comments under weaknesses. One additional suggestion: the authors found that pLLDT is predictive of flexible residues. Can they try to find AFm features that are predictive of the interface site? Such information may guide their docking to a local site.

This is a great idea that we and others have been thinking about considerably. Prior work by Burke et al. (Towards a structurally resolved human protein interaction network) examines AlphaFold’s ability to predict PPIs. For high-confidence predicted models of interacting protein complexes, the authors showed that pDockQ correlated reasonably well with correct protein interactions.

That being said, binding site identification, particularly in a partner-agnostic fashion, i.e. determining binding patches on a given protein, is an area of on-going research . We hope a future study examines AlphaFold3 or ESM3 specifically for this task.

“Further, we tested multiple thresholds to estimate the optimum cut-off for distinguishing near-native structures (defined as an interface-RMSD < 4 Å) from the predictions. Figure 3.B summarizes the performance with a confusion matrix for the chosen interface-pLDDT cutoff of 85. 79 % of the targets are classified accurately with a precision of 75%, thereby validating the utility of interface-pLDDT as a discriminating metric to rank the docking quality of the AFm complex structure predictions. With AlphaFold3 and ESM3 being released, investigating features that could predict flexible residues or interface site would be valuable, as this information may guide local docking.”

Minor:Page 3, lines 73-77, state how many targets were curated from DB5.5.

We have now clarified this in the manuscript. All 254 targets curated from DB5.5 at the time of this benchmark study.

“For each protein target, we extracted the amino acid sequences from the bound structure and predicted a corresponding three-dimensional complex structure with the ColabFold implementation of the AlphaFold multimer v2.3.0 (released in March 2023) for the 254 benchmark targets from DB5.5.”

In Figure 1, the color used for medium is too difficult to distinguish from the grey color used for rigid.

We thank you for this suggestion. We have updated the color to olive. Further, based on Reviewer 2’s suggestions, we have moved this plot to the Supplementary.

**Reviewer #2 (Public Review):**
Summary:In short, this paper uses a previously published method, ReplicaDock, to improve predictions from AlphaFold-multimer. The method generated about 25% more acceptable predictions than AFm, but more important is improving an Antibody-antigen set, where more than 50% of the models become improved.When looking at the results in more detail, it is clear that for the models where the AFm models are good, the improvement is modest (or not at all). See, for instance, the blue dots in Figure 6. However, in the cases where AFm fails, the improvement is substantial (red dots in Figure 6), but no models reach a very high accuracy (Fnat ~0.5 compared to 0.8 for the good AFm models). So the paper could be summarized by claiming, "We apply ReplicaDock when AFm fails", instead of trying to sell the paper as an utterly novel pipeline. I must also say that I am surprised by the excellent performance of ReplicaDock - it seems to be a significant step ahead of other (not AlphaFold) docking methods, and from reading the original paper, that was unclear. Having a better benchmark of it alone (without AFm) would be very interesting.

We thank the reviewer for highlighting the performance of ReplicaDock. ReplicaDock alone is benchmarked in the original paper (10.1371/journal.pcbi.1010124), with full details on the 2022 version of DB5.5 in the supplement. Indeed ReplicaDock2 achieves the highest reported success rates on flexible docking targets reported in the literature (until this AlphaRED paper!).

Regarding this statement about “the paper could be summarized…” it might be helpful to give more context. ReplicaDock is a replica exchange Monte Carlo sampling approach for protein docking that incorporates flexibility in an induced-fit fashion. However, the choice of which backbone residues to move is solely dependent on contacts made during each docking trajectory. In the last section of the ReplicaDock paper, we introduced “Directed Induced-fit” where we biased the backbone sampling only towards those residues where we knew the backbone is flexible (this information is obtained because for the benchmark set, we had both unbound and bound structures and hence could cherry-pick the specific residues which are mobile). We agree with the reviewers that AlphaRED is essentially a derivative of ReplicaDock, however, the two major claims that we make in this paper are:

(1) AlphaFold pLDDT is an effective predictor of backbone flexibility for practical use in docking.

(2) We can automate the Directed InducedFit approach within ReplicaDock by utilizing this pLDDT information per residue for conformational sampling in protein docking; and in doing so, create a pipeline that would allow us to go from sequence-to-structure-to-complex, specifically capturing conformational changes.

To conclude these claims, we pose the following questions in the Introduction:

“(1) Do the residue-specific estimates from AF/AFm relate to potential metrics demonstrating conformational flexibility?

(2) Can AF/AFm metrics deduce information about docking accuracy?

(3) Can we create a docking pipeline for in-silico complex structure prediction incorporating AFm to convert sequence-to-structure-to-docked complexes?”

This work requires a pipeline, the center of which lies in ReplicaDock as a docking method, but has functionalities that were absent in prior work. The goal is also to develop a one-stop shop without manual intervention (a prerequisite for biasing backbone sampling in ReplicaDock) that could be utilized by structural biologists efficiently.

We clarify this points in the abstract and main text as follows:

Abstract: “In this work, we combine AlphaFold as a structural template generator with a physics-based replica exchange docking algorithm to better sample conformational changes.”

Introduction:

“The overarching goal is to create a one-stop, fully-automated pipeline for simple, reproducible, and accurate modeling of protein complexes. We investigate the aforementioned questions and create a protocol to resolve AFm failures and capture binding-induced conformational changes. We first assess the utility of AFm confidence metrics to detect conformational flexibility and binding site confidence.”

These results also highlight several questions I try to describe in the weakness section below. In short, they boil down to the fact that the authors must show how good/bad ReplicaDock is at all targets not only the ones where AFm fails. In addition, I have several more technical comments.Strengths:Impressive increase in performance on AB-AG set (although a small set and no proteins).

We thank the reviewer for their comments.

Weaknesses:The presentation is a bit hard to follow. The authors mix several measures (Fnat, iRMS, RMSDbound, etc). In addition, it is not always clear what is shown. For instance, in Figure 1, is the RMSD calculated for a single chain or the entire protein? I would suggest that the author replace all these measures with two: TM-score when evaluating the quality of a single chain and DockQ when evaluating the results for docking. This would provide a clearer picture of the performance. This applies to most figures and tables.

We apologize for the lack of clarity owing to different metrics. Irms and fnat are standard performance metrics in the docking field, but we agree that DockQ would be simpler when the detail of the other metrics are not required. We have updated the figures Figure 5 and Figure 8 to also show DockQ comparisons.

Regarding Figure 1, as highlighted in Line 90 of the main-text, “Figure 1 shows the Ca-RMSD of all protein partners of the AFm predicted complex structures with respect to the bound and the unbound.” As suggested by the reviewer in their further comments, we have moved this FIgure to the Supplementary. We have also included TM-score comparison in the Supplementary (SupFig S2) and included clarifying statements in the main text:

“We also tested TM-scores to measure the structural deviations of the AFm predicted complex structures with respect to the bound and unbound structures (Supplementary Figure S2). However, this metric is not sensitive enough to detect the subtle, local conformational changes upon binding.”

For instance, Figure 9 could be shown as a distribution of DockQ scores.

We have now updated Figure 5 to include DockQ scores in Panel D. Since DockQ is a function of iRMSD, fnat and L-RMSD, it shows cumulative improvement in performance. Some of the nuanced details, such as, the protocol improves i-RMSD considerably but fnat improvement is lacking, and can highlight whether backbone sampling is the challenge or is it sidechain refinement.Therefore, we need to retain the iRMSD and fnat metrics in panel A-C . But We have incorporated this in the main text as follows:

“Finally, to evaluate docking success rates, we calculate DockQ for top predictions from AFm and AlphaRED respectively (Figure 5D). AlphaRED demonstrates a success rate (DockQ>0.23) for 63% of the benchmark targets. Particularly for Ab-Ag complexes, AFm predicted acceptable or better quality docked structures in only 20% of the 67 targets. In contrast, the AlphaRED pipeline succeeds in 43% of the targets, a significant improvement.”

Further, we have reevaluated success rates in Figure 8 (previously Figure 9) and have updated the manuscript to report these updated success rates.

“By utilizing the AlphaRED strategy, we show that failure cases in AFm predicted models are improved for all targets (lower Irms for 97 of 254 failed targets) with CAPRI acceptable-quality or better models generated for 62% of targets overall (Fig 8)”.

The improvements on the models where AFm is good are minimal (if at all), and it is unclear how global docking would perform on these targets, nor exactly why the plDDT<0.85 cutoff was chosen.

We agree with the reviewers that the improvement on the models with good AFm predictions is minimal. We acknowledge this in the text now as follows:

“Most of the improvements in the success rates are for cases where AFm predictions are worse. For targets with good AFm predictions, AlphaRED refinement results in minimal improvements in docking accuracy.”

The choice of pLDDT cutoff = 85 is elaborated in the “Interface-pLDDT correlates with DockQ and discriminates poorly docked structures” section, paragraph 3. Briefly, we tested multiple metrics and the interface pLDDT had the highest AUC, indicating that it is the best metric for this task. For interface-pLDDT we tested multiple thresholds, and the cutoff of 85 resulted in the highest percentage of true-positive and true-negative rates. This is illustrated with the confusion matrix in Figure 3.B with the precision scores. We now clarify this in the text as follows:

“With interface-pLDDT as a discriminating metric, we tested multiple thresholds to estimate the optimum cut-off for distinguishing near-native structures (defined as an interface-RMSD < 4 Å) from the predictions. Figure 3B summarizes the performance with a confusion matrix for the chosen interface-pLDDT cutoff of 85. 79% of the targets are classified accurately with a precision of 75%, thereby validating the utility of interface-pLDDT as a discriminating metric to rank the docking quality of the AFm complex structure predictions.”

To better understand the performance of ReplicaDock, the authors should therefore (i) run global and local docking on all targets and report the results, (ii) report the results if AlphaFold (not multimer) models of the chains were used as input to ReplicaDock (I would assume it is similar). These models can be downloaded from AlphaFoldDB.

The performance of ReplicaDock on DB5.5 is tabulated in our prior work (https://doi.org/10.1371/journal.pcbi.1010124) and we direct the reviewers there for the detailed performance and results. In our opinion, the benchmark suggested by the reviewer would be redundant and not worth the computational expense.

The scope of this paper is to highlight a structure prediction + physics-based modeling pipeline for docking to adapt to the accuracy of up-and-coming structure prediction tools.

Using AlphaFold monomer chains as input and benchmarking on that, albeit interesting scientifically, will not be useful for either the pipeline or biologists who would want a complex structure prediction. We thank the authors for their comments but want to reemphasize that the end goal of this work is to increase the accuracy of complex structure predictions and PPIs obtained from computational tools.

Further, it would be interesting to see if ReplicaDock could be combined with AFsample (or any other model to generate structural diversity) to improve performance further.

We would like to highlight that ReplicaDock is a stand-alone tool for protein docking and here we demonstrate the ability of adapting it with metrics derived from AlphaFold or other structure prediction tools (say ESMFold) such as pLDDT for conformational sampling and improving docking accuracy. We definitely agree that adapting it to use with tools such as AFSample will be interesting but it is out of scope of this work.

The estimates of computing costs for the AFsample are incorrect (check what is presented in their paper). What are the computational costs for RepliaDock global docking?

The authors of the AFSample paper report that “AFsample requires more computational time than AF2, as it generates 240 models, and including the extra recycles, the overall timing is 1000 more costly than the baseline.” We have reported these exact numbers in our manuscript.

The computational costs of ReplicaDock are 8-72 CPU hours on a single node with 24 processors as reported in our prior work.

For AlphaRED, the costs are slightly higher owing to the structure prediction module in the beginning and are up to 100 CPU hrs for our largest (max Nres) target.

It is unclear strictly what sequences were used as input to the modelling. The authors should use full-length UniProt sequences if they were not done.

We report this in the methods section of the manuscript as well as in Figure 5. Full length complex sequences were used for the models that we extracted from DB5.5.

“As illustrated in Fig. 5, given a sequence of a protein complex, we use the ColabFold implementation of AF2-multimer to obtain a predictive template.”

We clarify this in the methods section as:

“For each target in the DB5.5 dataset, we first extracted the corresponding FASTA sequence for the bound complex and then obtained AlphaFold predicted models with the ColabFold v1.5.2 implementation of AlphaFold and AlphaFold-multimer (v.2.3.0).”

The antibody-antigen dataset is small. It could easily be expanded to thousands of proteins. It would be interesting to know the performance of ReplicaDock on a more extensive set of Antibodies and nanobodies.

This work demonstrates the performance on the docking benchmark, i.e. given unbound structure can you predict the bound complexes. With this regard, our analysis has been focussed on targets where both the unbound and bound structures are available so that we could evaluate the ability of AlphaRED on modeling protein flexibility and docking accuracy. For antibody-antigen complexes, there are *only 67 structures* with both unbound and bound complexes available and they constituted our dataset. Benchmarking AlphaRED on all antibody-antigen targets can give biased results as most Ab-Ag complexes are in AlphaFold training set. Further, our work is more aimed towards predicting conformational flexibility in docking and not rigid-body docked complexes, so benchmarking on existing bound Ab-Ag structures is out of scope for this work.

Using pLDDT on the interface region to identify good/bas models is likely suboptimal. It was acceptable (as a part of the score) for AlphaFold-2.0 (monomer), but AFm behaves differently. Here, AFm provides a direct score to evaluate the quality of the interaction (ipTM or Ranking Confidence). The authors should use these to separate good/bad models (for global/local docking), or at least show that these scores are less good than the one they used.

We thank the reviewers for this suggestion.

**Reviewer #2 (Recommendations For The Authors):**
Some Figures could be skipped/improvedFig 1: Use TM-score instead a much better measure (and the figure is not necessary).

Figure 1 compares the bias of AlphaFold towards unbound or bound forms of the proteins. We believe that this figure highlights the slight inherent bias of AlphaFold towards bound structures over unbound.

As the reviewers have suggested we have included a plot comparing the TM-scores for the structures. Further, we have moved this figure to the Supplementary.

Fig 2. Skip B (why compare RMSD with pLDDT?). Add a figure to see how this correlates over all targets not just two.

RMSD and LDDT both represent metrics to evaluate conformational variability between two structures, such as the bound and unbound forms of the same protein structure. On one hand where RMSD measures overall deviation of residues, LDDT allows the estimation of relative domain orientations and concerted proteins. We have elaborated this in Methods as well as in the Results section titled “AlphaFold pLDDT provides a predictive confidence measure for backbone flexibility”.

The data for the benchmark targets is now included in the Supplementary (Supplementary Figures S3-S4).

Fig 3. Color the different chains of a protein differently. Thereby the Receptor/Ligand/Bound labels can be omitted.

We thank the reviewers for this suggestion. However, the color scheme is chosen to highlight (1) the relative orientation of protein partners relative to each other. We have ensured that the alignment is over one partner (Receptor) so that you could see the relative orientation of the other partner (Ligand) in the modeled protein over the bound structure (in one color). (2) The coloring of the receptor and ligand chain is by pLDDT (from red to blue) to highlight that for decoys with incorrectly predicted interfaces, the pLDDT scores of the interface residues are indeed lower and can be a discriminating metric. We elaborate this in the caption of Figure 3 as well as in the section “Interface-pLDDT correlates with DockQ and discriminates poorly docked structures”. Coloring the chains of a protein differently will obfuscate the point that we are aiming to make and will be inconclusive for the readers as they would need to rely only on quantitative metrics (Irms and DockQ) reported but won’t be able to visualize the interface pLDDT of the incorrectly bound structures. We hope that this justifies the choice of our color scheme.

Fig 4. Include RankConf, ipTM, pDockQ, and other measures in the plos (they are likely better). Include DockQ for the top targets. It is difficult to estimate for multi chain complexes.

We thank the reviewer for this suggestion. We have now included the DockQ performances for all targets in Figure 5 (previously Figure 6) as well as re-evaluated our final success rates based on the DockQ calculations in Figure 8 (previously Figure 9).

Fig 5. use a better measure to split (see above).

We have elaborated on the choice of the split for the comments above and the interface pLDDT threshold of 85 is a decision made post observation on the docking benchmark. We do want to highlight that the cut-off is arbitrary and in our online server (ROSIE) as well as in custom scripts, this cut-off can be tuned by the user as required. We would suggest a cut-off of 85 based on our observations but the users are welcome to tune this as per their needs.

Fig 6. Replace lrms/fnat with DockQ.

We have now included DockQ scores in our manuscript.

Fig 7. Color the different chains of a protein differently.

We have colored the protein chains differently. AlphaFold models are in Orange, Bound complexes are in Gray, and predicted proteins from AlphaRED are in Blue-Green indicating the two partners. All models are aligned over the receptor so relative orientations of the ligand protein can be observed.

Fig 8 Color the different chains of a protein differently.

The chains are colored differently. We would like the reviewer to elaborate more on what they would like to observe as we believe our color scheme makes intuitive sense for readers.

Fig 9. Use DockQ instead of CAPRI criteria.

The figure has been updated based on DockQ. To elaborate, the CAPRI criteria is set based on DockQ scores as elaborated in the figure caption.